# The Central Role of Macrophages in Long COVID Pathophysiology

**DOI:** 10.3390/ijms27010313

**Published:** 2025-12-27

**Authors:** Philip Mcmillan, Anthony J. Turner, Bruce D. Uhal

**Affiliations:** 1McMillan Research Ltd., 71-75 Shelton Street, Covent Garden, London WC2H 9JQ, UK; philip.mcmillan@nhs.net; 2Faculty of Biological Sciences, School of Biomedical Sciences, University of Leeds, Leeds LS2 9JT, UK; a.j.turner@leeds.ac.uk; 3Department of Physiology, Michigan State University, East Lansing, MI 48824, USA

**Keywords:** Long COVID, PASC, macrophage, MAIT cells, neuroinflammation

## Abstract

This review article attempts to provide a unifying hypothesis to explain the myriad of symptoms and predispositions underlying the development of PASC (Postacute Sequelae of COVID), often referred to as Long COVID. The hypothesis described here proposes that Long COVID is best understood as a disorder of persistent immune dysregulation, with chronic macrophage activation representing the fundamental underlying pathophysiology. Unlike transient post-viral syndromes, Long COVID involves a sustained innate immune response, particularly within monocyte-derived macrophages, driven by persistent spike protein (peripherally in MAIT cells and centrally in Microglial cells), epigenetic imprinting, and gut-related viral reservoirs. These macrophages are not merely activated temporarily but also become epigenetically “trained” into a prolonged inflammatory state, as demonstrated by enduring histone acetylation markers such as H3K27acDNA Reprogramming. It is proposed that recognizing macrophage activation as the central axis of Long COVID pathology offers a framework for personalized risk assessment, targeted intervention, and therapeutic recalibration.

## 1. Introduction

Long COVID, or post-acute sequelae of SARS-CoV-2 infection (PASC), has emerged as a complex syndrome characterized by persistent or relapsing symptoms beyond the resolution of acute infection that affects an estimated 10–30% of all infected individuals [1]. Long COVID presents with a wide range of symptoms, including fatigue, cognitive dysfunction, breathlessness, dysautonomia, chest pain, and myalgias. This broad array of complaints bears striking similarity to other post-infectious conditions such as myalgic encephalomyelitis/chronic fatigue syndrome (ME/CFS), fibromyalgia, and chronic Lyme disease [2,3,4], suggesting shared pathophysiological mechanisms rooted in immune dysregulation rather than organ-specific injury.

A crucial distinction must be made between hyperactive immune-mediated cytokine storm organ damage resulting from severe acute COVID-19 versus more chronic immune-mediated dysfunctions observed after mild or non-hospitalized infection [5,6]. In severe cases, SARS-CoV-2 can cause direct viral and immune-mediated cytopathic injury and thromboinflammation, leading to complications such as pulmonary fibrosis, myocarditis, acute kidney injury, and cerebrovascular damage [7,8]. These sequelae, often verified by imaging or biomarkers, reflect structural or ischemic tissue damage. Roughly 10% of patients who suffered severe COVID reported persistent symptoms for 12 weeks or longer after the initial infection [5]. However, in many Long COVID patients who had only mild initial illness, which by various estimates can comprise from 10 to 80% of those with mild initial illness [5,9], there is no radiological or laboratory evidence of organ pathology. Yet, symptoms can persist for months, implying alternative pathophysiological mechanisms [9]. Multi-omic studies consistently show that Long COVID after mild acute disease is driven primarily by persistent innate immune activation, especially monocyte/macrophage dysregulation [10].

Recent evidence suggests that immune cell persistence and reprogramming play a central role in these non-organ-damaged-related syndromes. Patterson et al. found that the SARS-CoV-2 S1 spike protein persisted in CD16^+^ monocytes up to 15 months post-infection in individuals with Long COVID, despite the absence of ongoing viral replication [11]. Complementary work by Simonis et al. revealed that mRNA vaccination induced durable epigenetic reprogramming in monocyte-derived macrophages, characterized by H3K27ac modifications and elevated IL-1β, TNF-α, and other inflammatory cytokines [12]. These findings demonstrate that spike protein exposure, whether through infection [13] or vaccination, can lead to long-lasting macrophage activation [14].

Furthermore, persistent spike protein has been detected in individuals with Long COVID-like symptoms following Covid vaccination, in the absence of nucleocapsid antibodies, indicating that similar immune imprinting may occur independently of natural infection [11]. In these individuals, S1 and S2 spike protein fragments were identified in CD16^+^ monocytes 245 days post-vaccination using mass spectrometry [15]. While immune dysregulation and organ damage can co-exist, particularly in those with severe acute disease, they must be distinguished in order to better understand disease manifestation. For example, structural lung fibrosis or myocarditis implies irreversible tissue remodeling, often requiring organ-specific management, whereas macrophage-driven immune dysfunction may respond to immunomodulation or metabolic correction. Confusing these entities could risk therapeutic misdirection and hinder accurate phenotyping of Long COVID.

This paper proposes an immune-centric framework for Long COVID, emphasizing macrophage activation, spike persistence, cytokine profiling, and epigenetic reprogramming as central themes. We propose that this model not only reconciles the multisystem nature of symptoms but also links Long COVID to a broader family of post-viral syndromes.

## 2. Macrophage Subsets and Their Role in Long COVID Pathophysiology

Macrophages play a central role in immune regulation, inflammation, and tissue repair, and exist in phenotypically distinct subtypes derived from circulating monocytes: classical (CD14^+^, CD16^−^), intermediate (CD14^+^, CD16^+^), and non-classical (CD14^lo^, CD16^+^). Intermediate and non-classical monocytes express high levels of CCR5 and CX3CR1, which promotes their migration to and retention within previously inflamed tissues and vasculature [11,15]. In Long COVID, macrophage hyperactivation appears to be a core mechanism of disease, especially in an environment where active inflammation is already present. Intermediate and non-classical monocytes have been shown to harbor SARS-CoV-2 S1 protein for up to 15 months post-infection, despite an absence of full-length viral RNA, indicating persistent antigenic stimulation [11]. These cells are associated with elevated levels of inflammatory cytokines such as IL-6, CCL5, and sCD40L, as well as platelet and endothelial activation, which are characteristic features of PASC [1]. SARS-CoV-2 spike protein further impairs lysosomal acidification and autophagosome–lysosome fusion in monocytes, reducing their ability to degrade internalized viral antigens [16]. This impairment is amplified in CD16^+^ non-classical monocytes, which inherently possess lower lysosomal enzyme content and diminished phagolysosomal activity, predisposing them to prolonged spike protein retention [17]. Further compounding this state is macrophage epigenetic reprogramming causing prolonged and sustained inflammation. SARS-CoV-2 mRNA vaccination induces persistent histone H3K27 acetylation in monocyte-derived macrophages, sustaining pro-inflammatory gene expression even months after exposure. This trained immunity effect enhances the production of cytokines like IL-1β and IL-18, reinforcing chronic immune activation and contributing to the Long COVID phenotype [12].

## 3. Macrophage Activation: A Common Pathophysiological Axis in Severe and Long COVID?

Severe COVID-19 and Long COVID appear as distinct clinical syndromes, yet they are connected through a shared immunological axis: the dysregulation of monocyte-macrophage populations. Evidence shows that innate immune subsets, especially CD14^+^ CD16^+^ inflammatory monocytes and CD16^+^ non-classical monocytes, play central roles in both the acute cytokine storm of severe disease and the chronic, smoldering inflammation of Long COVID. In severe COVID-19, a delayed Type I interferon (IFN-I) response precipitates a wave of inflammatory monocyte-macrophages (IMMs) into the lungs [18]. These IMMs secrete TNF, IL-6, and IL-1β, which damage vascular endothelium, cause alveolar leakage, and suppress adaptive T-cell responses. The profound T-cell lymphopenia seen in severe COVID-19 may arise from hyperactivated myeloid-derived suppressor cells (MDSCs) driving destructive inflammation within secondary lymphoid organs, thereby collapsing normal T-cell development and survival [19]. Notably, in murine models, depletion of IMMs or ablation of IFN-I signaling prevented death despite unchanged viral loads, indicating that immune-mediated damage, not viral replication, is likely to be the primary driver of pathology in severe disease [18].

In contrast, Long COVID after mild infection reflects a different trajectory of the same immune cell populations. Rather than rapid recruitment in the lungs, infected monocytes persist in a chronically activated or programmed state. Patterson et al. [11] identified the SARS-CoV-2 S1 protein persisting in CD14^−^ CD16^+^ non-classical monocytes in patients with long COVID symptoms—up to 15 months post-infection. These monocytes were also associated with cytokine profiles indicative of ongoing immune activation, including elevated sCD40L and IL-6 [1]. Crucially, Simonis et al. [12] have demonstrated that even in the absence of infection, exposure to SARS-CoV-2 mRNA via vaccination induces durable epigenetic reprogramming of monocyte-derived macrophages. Through persistent H3K27ac acetylation, key immune genes such as *IL1B*, *SYK*, and *NOD2* remain transcriptionally poised, leading to a memory-like pro-inflammatory response upon restimulation. This form of “trained immunity” mimics features seen in Long COVID, supporting a shared mechanism.

Further reinforcing this paradigm, S1 protein has also been identified in CD16^+^ monocytes of individuals with Long COVID-like symptoms following COVID-19 vaccination [14]—despite no history of SARS-CoV-2 infection. These cells exhibited the same inflammatory cytokine signatures (IL-6, IL-8, CCL5, sCD40L) and were misclassified as long COVID patients by immune-based machine learning algorithms [1]. Collectively, these findings support a unified hypothesis: both severe and Long COVID reflect divergent expressions of a common immunopathology rooted in macrophage dysfunction. In severe cases, it manifests as acute immunopathology. In Long COVID, it persists as chronic, low-grade inflammation—different timelines, but immunologically linked through the same innate effector axis.

## 4. Aortic Inflammation and Vasa Vasorum Pathology in Long COVID

Recent evidence suggests that Long COVID chest pain may arise not from classical coronary artery obstruction, but from chronic inflammation of the aortic arterial wall and its vasa vasorum. Positron emission tomography (PET) imaging studies in patients with acute and post-acute COVID have identified increased labelled glucose uptake, indicating rising metabolic activity in the thoracic aorta, even in individuals without overt cardiovascular disease [20]. This finding aligns with autopsy observations of mononuclear infiltrates within the adventitia and perivascular lymphocytic cuffing around vasa vasorum, features typical of immune-mediated vascular injury rather than atherosclerosis [21,22].

The mechanism appears to indicate monocyte/macrophage persistence and spike protein localization in vascular niches. Non-classical CD16^+^ monocytes, already implicated in Long COVID [1,11], traffic along the CX3CR1–fractalkine axis to inflamed endothelium in the vasa vasorum and penetrate the adventitial microvasculature. This may cause an inflammatory loop through IL-6, TNF-α, and tissue factor expression [12]. The consequence is microvascular inflammation, hypoxia of the aortic media, and compensatory fibrosis, which together accelerate arterial stiffening. This aligns with the CARTESIAN study’s observation of rapid increases in pulse wave velocity (PWV) post-COVID, equivalent to adding 5–7 years of vascular age within months [21].

A fundamental requirement for recovery from many inflammatory diseases is not merely suppression of initial immune activation or infection, but successful resolution driven in large part by macrophages [23]. During acute infection, inflammatory monocytes and macrophages adopt a pro-inflammatory state that supports pathogen control, but tissue repair requires a subsequent shift toward a pro-resolving/regulatory phenotype characterized by reduced IL-1β/TNF signaling, increased IL-10/TGF-β tone, enhanced mitochondrial oxidative metabolism with efficient clearance of cellular debris and apoptotic cells [24]. Failure of this switch is increasingly recognized in SARS-CoV-2 infection, specifically, efferocytosis of infected apoptotic cells can become dysfunctional and paradoxically suppress macrophage anti-inflammatory programming, impairing tissue repair and perpetuating inflammatory signaling [25].

Persistent macrophage inflammation can lead to tissue cellular changes that may lead to extracellular matrix remodeling. In the context of Long COVID, macrophages may continue to secrete profibrotic mediators, which could lead to excessive collagen deposition [26]. This has been demonstrated with post-viral SARS-CoV-2 lung involvement driven primarily by macrophage-induced fibrosis [27]. Although lungs would be the most likely location, macrophage-induced fibrosis could also occur in the vascular system [28]. These processes provide a mechanistic framework linking macrophage persistence in Long COVID to observed vascular stiffening, pulmonary fibrosis, and organ-specific functional decline, even in the absence of ongoing viral replication or acute tissue injury.

## 5. Mait Cells, Gut Barrier Dysfunction, and Systemic Inflammation in Long COVID

SARS-CoV-2 infection frequently initiates in the upper respiratory tract, spreads to the lungs, and rapidly disseminates via the bloodstream to distal organs, including the gastrointestinal tract, where intestinal epithelial cells expressing the coronaviral receptor angiotensin-converting enzyme-2 (ACE2) facilitate viral entry. The resulting disruption to gut homeostasis may lead to microbiome dysbiosis, characterized by an overgrowth of Gram-negative bacteria such as *Clostridia*, and depletion of commensals like *Bifidobacterium* and *Lactobacillus* [29]. This imbalance, coupled with increased intestinal permeability, permits microbial translocation and promotes chronic systemic inflammation. Mucosal-associated invariant T (MAIT) cells are a subset of innate-like T cells concentrated at mucosal barriers, particularly in the gut [30]. Although MAIT cells do not support productive replication of SARS-CoV-2, they can become activated by inflammatory cytokines and exposure to viral antigens, including spike protein, potentially delivered via exosomes released from infected epithelial or immune cells [31,32]. Post-infection, MAIT cells are highly sensitive to microbial metabolites and inflammatory cues, driving robust secretion of interferon-gamma (IFN-γ), which could contribute to sustained inflammation [30]. This scenario mirrors the chronic interferon signaling and epigenetic activation seen in monocyte-derived macrophages following SARS-CoV-2 mRNA vaccination [12].

Overactivation of MAIT cells extends beyond localized mucosal sites, as these cells traffic via lymphatic channels and secrete cytokines systemically, linking gut barrier breakdown to distant organ dysfunction [33,34]. Their trafficking is governed primarily by chemokine receptors such as CCR6, CXCR6, and CCR5, which facilitate homing to mucosal and vascular niches, expressing ligands including CXCL16 and CCL20, particularly under inflammatory conditions [35]. SARS-CoV-2–associated inflammation promotes systemic MAIT cell activation [30] through cytokine-driven, TCR-independent mechanisms—most notably IL-12, IL-18, and type I interferons—leading to widespread IFN-γ and TNF-α release [36]. The capacity for mucosal activation followed by systemic redistribution, including the central nervous system [37], positions MAIT cells as a critical immunological bridge linking gut barrier dysfunction to persistent macrophage-driven inflammation in long COVID. This may help explain the broad symptomatology of Long COVID, including persistent fatigue, neurocognitive impairment, and cardiovascular abnormalities [34]. Moreover, IFN-γ produced by MAIT cells upregulates endothelial fractalkine and its receptor CX3CR1, promoting the survival and recruitment of non-classical CD16^+^ monocytes, which have been shown to harbor persistent spike protein months after infection or vaccination [11,12].

The immune dysregulation observed in long COVID extends beyond mucosal inflammation and implicates systemic innate immune pathways, particularly the interaction between MAIT cells and monocyte-derived macrophages. MAIT cells are known to express high levels of pro-inflammatory cytokines such as interferon-gamma (IFN-γ), tumor necrosis factor-alpha (TNF-α), and interleukin-17 (IL-17) upon activation, all of which can profoundly alter macrophage polarization and function [37]. In the context of chronic MAIT cell activation through associated intestinal inflammation [34], as seen in post-COVID states, this cytokine milieu sustains macrophages in a pro-inflammatory state, suppressing resolution and promoting ongoing tissue inflammation and damage. Macrophages exist in a spectrum between classically activated (M1) pro-inflammatory phenotypes and alternatively activated (M2) reparative states [37]. In Long COVID, this polarization is often compounded by underlying histone modifications and epigenetic programming changes that impair macrophage plasticity, rendering them unable to revert to a regulatory phenotype. This leads to sustained immune activation, tissue injury, and fibrosis in affected organs, including the gut, lungs, and vascular system [7].

Further complicating this interaction, activated MAIT cells upregulate CD40L (CD154), a key ligand that interacts with CD40 on macrophages and dendritic cells. This CD40–CD40L engagement promotes further maturation and activation of antigen-presenting cells, amplifying the inflammatory loop [38]. Additionally, MAIT cell–derived exosomes may carry RNA fragments and viral antigens that continue to stimulate macrophages via toll-like receptors (TLRs), potentially sustaining type I and II interferon signaling [32]. This contributes to a chronic, unresolved immune response marked by features of immune exhaustion in some cell subsets, while simultaneously preserving high inflammatory tone. We hypothesize that together, this maladaptive MAIT–macrophage axis underpins much of the immune dysfunction in Long COVID. The sustained cross-activation between these two innate-like populations can explain the persistence of systemic symptoms and inflammatory biomarkers long after viral clearance. Interrupting this feedback loop, through modulation of MAIT cell responses, IFN-γ signaling blockade, or epigenetic reprogramming of macrophages, represents a promising avenue for therapeutic intervention and immune restoration in post-COVID conditions.

In accord with this viewpoint, the manipulation of MAIT cell responses could offer a promising therapeutic avenue to help downregulate macrophage immune dysregulation. Interventions aimed at restoring gut microbial balance, suppressing excessive IFN-γ signaling, or stabilizing the epithelial barrier may interrupt this pathogenic cycle. We suggest that future research should explore dietary, microbiome-based, and immunomodulatory strategies to regulate MAIT cell activity in Long COVID and vaccine-associated syndromes [31].

## 6. Cd8^+^ T Cell Hyperactivation, Macrophage Activation and Epigenetic Memory

Effector memory (Tem) and tissue-resident memory (Trm) CD8^+^ T-cell populations represent functionally distinct but phenotypically overlapping subsets whose identification is influenced by gating strategy and tissue context [39] and could influence macrophage activation and impaired immune resolution even after clearance of replicative virus. The study by Renner et al. [40] revealed a striking hyper-reactivity of CD8^+^ T cells, particularly effector memory (Tem) and tissue-resident memory (Trm) subsets, together with significant IL-3 production in patients recovering from mild COVID-19. This elevation was notably persistent over time and associated with prolonged symptoms resembling Long COVID [41]. Interleukin-3, a hematopoietic cytokine, is known for its role in recruiting and stimulating myeloid cells, especially monocytes and macrophages. The unique upregulation of IL-3 within CD8^+^ T cells post-COVID implies a potential mechanism by which T cell dysfunction drives chronic myeloid activation even after viral clearance.

This T cell-driven IL-3 release intersects mechanistically with emerging evidence of macrophage epigenetic reprogramming post-mRNA vaccination. The work of Simonis et al. [12] documents persistent histone modifications in monocyte-derived macrophages after mRNA vaccination, locking these cells into a pro-inflammatory state via H3K27 acetylation—particularly in the absence of traditional adaptive memory responses. This provides a plausible mechanism for how macrophages, once stimulated by T cell-derived IL-3, may remain in an activated phenotype long after the resolution of acute infection or immunization.

MAIT cells, which are enriched at mucosal sites and capable of rapid cytokine release, are also elevated in Long COVID and ME/CFS [3]. While the featured IL-3 secreting T cells are largely conventional CD8^+^, the possibility that MAIT cells contribute synergistically cannot be excluded, especially as MAIT cells can enhance myeloid responses via IL-17 and IFN-γ, and potentially induce IL-3 under non-classical activation states [30]. Their interaction with tissue macrophages, especially in lung and gut mucosa, further supports the model where chronic mucosal immune activation sustains systemic macrophage dysregulation [42].

## 7. Clinical Implications: Brainstem Microglial Activation and Chronic Symptoms

The downstream effects of this macrophage overstimulation extend into the central nervous system. In the context of Long COVID, this suggests that T cell–derived IL-3 could not only drive peripheral macrophage inflammation but also microglial activation in vulnerable CNS regions like the brainstem, an area implicated in autonomic dysfunction, fatigue, and cognitive impairment [43]. The convergence of persistent spike protein in monocytes, epigenetic macrophage training [11,37], and IL-3–driven neuroimmune signaling [40] could create a “feed-forward loop” which, at least theoretically, could sustain long COVID pathology.

This chronic spike protein antigenic stimulation may be further compounded by epigenetic reprogramming in activated macrophages after vaccination [12]. These changes could prime macrophages for heightened inflammatory responses to both SARS-CoV-2–related and unrelated pathogen-associated molecular patterns (PAMPs), demonstrating a state of trained innate immunity that persists for months beyond antigen exposure [14]. This supports the hypothesis that Long COVID is driven not only by residual antigens but also by an immune system epigenetically primed for exaggerated responses.

In parallel, the gut emerges as a critical site for persistent antigen stimulation. We recently reviewed data showing the detection of SARS-CoV-2 spike protein and RNA in CD8^+^ T cells within the intestinal mucosa, particularly in patients with prior inflammatory bowel disease (IBD) [34]. MAIT cells, which bridge innate and adaptive immunity, may become dysregulated in this context, promoting systemic inflammation and contributing to the mosaic presentation of Long COVID symptoms.

Emerging research suggests that the epipharynx, a mucosal site rich in immune surveillance, may serve as an underrecognized nidus for chronic inflammation in Long COVID and related syndromes [44]. Persistent viral antigens, including spike protein RNA, have been identified in the epipharyngeal mucosa, where they may drive ongoing immune activation [45]. This localized inflammation may promote activation of mucosal macrophages and MAIT cells, contributing to systemic cytokine release and vascular dysregulation. Given the close anatomical proximity between the inflamed epipharynx and the nodose ganglion of the vagus nerve, this relationship could help explain the vagal inflammation and autonomic dysfunction frequently reported in Long COVID, including orthostatic intolerance, dysautonomia, chronic cough, gastrointestinal disturbances, and altered respiratory drive [46,47,48].

Together, these findings support a model where immune dysfunction in Long COVID arises from a feedback loop of persistent spike antigen exposure, macrophage epigenetic reprogramming, and mucosal T-cell activation. The convergence of these mechanisms, particularly within the gut, may explain the chronic and relapsing nature of symptoms observed in affected individuals. Therapeutic strategies aimed at resolving viral reservoirs, reversing epigenetic activation in macrophages, and restoring mucosal immune tolerance may offer promising avenues for recovery.

## 8. Regulatory T-Cell Dysfunction as a Catalyst for Macrophage Overactivation in Long COVID

A key immunological axis in the pathogenesis of Long COVID centers on the dysfunction of regulatory T cells (Tregs), whose role is to restrain excessive immune responses and preserve immune homeostasis. One such mechanism involves the viral spike protein’s interaction with neuropilin-1 (NRP1), a key surface molecule required for Treg stability and suppressive function. Inflammatory cytokines like IL-6 and IFN-γ, highly expressed during acute infection or post-vaccine immune activation, further skew differentiation away from the regulatory lineage and toward pro-inflammatory Th17 cells, exacerbating immune imbalance [49,50]. As a result, regulatory control over innate immune cells, particularly macrophages, is weakened.

In this context of compromised Treg regulation, macrophages, especially of the M1 phenotype, are left unchecked. Both infection and vaccination can epigenetically reprogram monocyte-derived macrophages through histone modifications (notably H3K27Ac), locking them in a persistent pro-inflammatory state that sustains the release of IL-1β, IL-6, and TNF-α for months after the initial stimulus [12,37]. This effect may be compounded by the direct engagement of Toll-like receptors (TLR2 and TLR4) by spike protein and the inflammatory properties of lipid nanoparticle adjuvants in mRNA platforms [51]. These reprogrammed macrophages display features of trained innate immunity, contributing to long-term tissue inflammation, lymphadenopathy, and systemic symptoms reminiscent of autoimmune conditions [52].

The resulting feedback loop between dysfunctional Tregs and overactive macrophages establishes a state of chronic immune activation. As shown in both experimental and observational studies, this loop impairs the resolution of inflammation across multiple organ systems. In intestinal tissues, already primed by local inflammation or microbiota dysbiosis, this dynamic may contribute to persistent viral antigen presentation, further exhausting immune control mechanisms and reinforcing macrophage activation. In the CNS, the same axis could underlie microglial dysfunction and explain neuroinflammatory symptoms such as “brain fog” and autonomic instability. Thus, targeting the Treg/macrophage axis [53] represents a promising therapeutic strategy to break this pathological cycle in long COVID.

## 9. Interferon-Driven Mait Cell Hyperactivation and Its Role in Chronic Inflammation

A hallmark of immune dysregulation in Long COVID is a hyperactive mucosal interferon response [54] that disproportionately activates MAIT cells. Therefore, in individuals predisposed to long COVID, either through genetic variants or prior mucosal inflammation, this pathway appears over-responsive. Clinical evidence supporting this comes from a study where patients treated with interferon-α for hepatitis C developed persistent fatigue with clinical and immunological parallels to ME/CFS, suggesting that exaggerated interferon signaling can induce chronic fatigue-like illness in susceptible individuals [55].

During SARS-CoV-2 infection, in predisposed individuals, this primed interferon landscape facilitates rapid and excessive MAIT cell activation, leading to the production of IL-17 and TNF-α and migration of MAIT cells into inflamed tissues. In the gut, this response compromises epithelial tight junctions, may lead to increased intestinal permeability (“leaky gut”) and microbial dysbiosis [56]. The influx of microbial components such as lipopolysaccharide (LPS) into the systemic circulation perpetuates monocyte and macrophage activation. Within the epipharynx, a similar inflammatory process potentially connected to bacterial biofilms may also occur. Chronic stimulation pushes MAIT cells into a state of functional exhaustion, characterized by reduced cytokine secretion and upregulation of inhibitory receptors like PD-L1 and TIM-3, which prevents mucosal healing [57] and perpetuates systemic inflammation [46], leading to functional immune exhaustion while paradoxically perpetuating low-grade inflammatory cytokine release [58] such as occurs in inflammatory bowel disease [59]. This dysfunctional immune loop, initiated by an interferon-sensitive mucosal immune circuit, converges on persistent macrophage activation, a central mechanism of Long COVID pathology. CD16^+^ non-classical monocytes harboring spike protein remain inflammatory for months post-infection [11], and their activation is likely sustained by continuous microbial translocation and unresolved mucosal immune exhaustion.

## 10. Macrophage-Driven Microclot Formation: A Central Mechanism in Long COVID Pathophysiology

In individuals with Long COVID, a hallmark finding is the persistence of SARS-CoV-2 spike protein, particularly the S1 subunit, within CD16^+^ monocytes months after acute infection or vaccination, even in the absence of detectable viral RNA [15]. These non-classical and intermediate monocyte subsets are known to patrol vascular endothelium and exhibit increased expression of adhesion molecules, such as CX3CR1. Persistent spike-containing monocytes demonstrate heightened secretion of pro-inflammatory cytokines (e.g., IL-6, IL-8) and platelet-activating factors (e.g., sCD40L), suggesting they may initiate endothelial activation and vascular inflammation. This chronic immune activation creates a systemic environment conducive to thrombogenesis [60].

Recent insights into macrophage biology reveal that, under sustained inflammatory stimulation, macrophages can release macrophage extracellular traps (METs)—DNA–protein webs resembling neutrophil extracellular traps (NETs) [61]. These METs act as scaffolds for platelet adhesion and coagulation factor activation, directly contributing to fibrin-rich microthrombi. In Long COVID, where macrophages are epigenetically reprogrammed into a persistent pro-inflammatory state by spike exposure or mRNA vaccination [12,14], METs may form aberrantly and chronically. These traps, together with monocyte-derived tissue factor and cytokines, propagate a cycle of thromboinflammation [62] and microvascular obstruction, particularly in vulnerable capillary beds such as the brainstem, retina, and lungs.

The presence of microclots resistant to fibrinolysis has been repeatedly observed in Long COVID patients [60] and shares characteristics with myalgic encephalomyelitis/chronic fatigue syndrome (ME/CFS) [3,4]. Proteomic analyses have demonstrated these clots contain misfolded fibrinogen with amyloid-like properties, potentially triggered by the oxidative environment and inflammatory cytokines released from macrophages. These clots occlude capillaries, reduce oxygen delivery, and impair cellular metabolism, offering a unifying explanation for the diverse symptoms of Long COVID such as fatigue, neurocognitive dysfunction (“brain fog”), and post-exertional malaise. Importantly, this pattern of microclotting appears disproportionately common in Long COVID compared to other viral infections, likely due to the unique interplay of spike persistence, macrophage dysregulation, and endothelial injury [4,63].

## 11. Mast Cell Activation in Long COVID

Mast cells, more commonly recognized for their role in allergic responses, have gained increasing attention as potential contributors to the complex immunopathology of Long COVID [64,65]. These innate immune cells, strategically positioned at mucosal barriers, perivascular sites, and neuroimmune interfaces, are uniquely poised to interact with macrophage-derived inflammatory mediators. In Long COVID, macrophage activation driven by persistent viral antigens—such as spike protein detected within CD16^+^ monocytes—produces a sustained release of cytokines including IL-1β, IL-6, and TNF-α [1]. These cytokines are well-established mast cell activators, and in individuals with pre-existing mast cell hyper-reactivity or mast cell activation syndrome (MCAS), they may lower the threshold for mast cell degranulation or cytokine release [65]. Thus, mast cell involvement may represent a secondary but clinically significant amplification loop in the broader macrophage-mediated immune dysregulation [53].

Although case–control studies have shown that classical mast cell degranulation markers such as tryptase (TPSB2) and carboxypeptidase A3 (CPA3) are not universally elevated in Long COVID [66], these findings do not exclude mast cell activation occurring through non-degranulating pathways or within tissue compartments. Mast cells in the brain—especially in regions such as the hypothalamus, hippocampus, and thalamus—may contribute to neurocognitive symptoms by releasing histamine, prostaglandins, and cytokines that disrupt blood–brain barrier integrity and affect neurotransmission [67]. Similarly, in the gut, mast cells closely interact with tissue-resident macrophages and the microbiome, potentially driving localized inflammation and systemic immune priming. Persistent spike protein in the gut has been associated with Long COVID symptoms and may engage both cell types in mucosal immune activation [34].

In this context, mast cells should be viewed not as primary initiators, but as downstream amplifiers within a macrophage-dominant model of Long COVID. Individuals with underlying mast cell sensitivity (whether genetic, allergic, or microbiome-driven) may experience symptom exacerbation through this macrophage–mast cell axis. This dual-pathway model offers a compelling explanation for the heterogeneity of Long COVID and may support the targeted use of mast cell stabilizers, antihistamines, and leukotriene receptor antagonists in a subset of patients [65,68]. Recognizing this interaction is essential for understanding disease progression, stratifying Long COVID phenotypes, and guiding more personalized treatment strategies.

## 12. Brainstem Macrophage Activation and Choroid Plexus Autoimmunity in Long COVID

The brainstem remains a central locus in Long COVID pathophysiology owing to its dense microglial and macrophage populations and critical autonomic nuclei (e.g., raphe nuclei, dorsal vagal complex, nucleus tractus solitarius). Microglia in these regions are capable of shifting to pro-inflammatory states in response to stimuli (e.g., hypercapnia) and can drive maladaptive plasticity in autonomic circuits [69,70]. Postmortem and imaging studies confirm regional microglial density and activation in brainstem autonomic centers [71]. Meanwhile, clinical studies document a high prevalence of orthostatic intolerance, POTS, and dysautonomia in Long COVID patients—symptoms that map directly to brainstem dysfunction [72,73]. Blitshteyn [74] further localizes neuroinflammatory signatures to the dorsal inferior medulla in POTS and Long COVID, linking symptom clusters to autonomic nuclei in this region.

Originally discovered as a cardioregulatory enzyme [75] and subsequently as a coronaviral receptor, ACE2 expression in the brainstem further supports this vulnerability. Although autoimmune mechanisms have been proposed to underlie its pathogenesis [34], the persistence of brainstem dysfunction in Long COVID is more consistent with a non-autoimmune mechanism—one in which immune cell reprogramming, chronic cytokine signaling, and microvascular stress together sustain a state of localized neuroinflammation and tissue injury [76].

Recent high-resolution 7T MRI using quantitative susceptibility mapping (QSM) has identified increased magnetic susceptibility in these brainstem areas, reflecting regional iron accumulation [2]. This is interpreted as a marker of chronic microglial and macrophage activation, possibly driven by mitochondrial dysfunction and ferroptotic cell death, an iron-dependent mechanism linked to oxidative stress and impaired cellular metabolism.

Importantly, these findings align not with classical autoimmunity but rather with persistent immune hyperactivation and trained innate immunity. Studies of post-COVID and post-vaccine immune programming have revealed epigenetic imprinting in monocyte-derived macrophages, notably persistent H3K27 acetylation, which locks these cells into a sustained inflammatory phenotype [12]. This epigenetic memory is particularly significant in long-lived tissue-resident macrophages, including those in the CNS.

In close anatomical proximity to the brain stem and other brain regions, the choroid plexus (CP) functions not only as a cerebrospinal fluid–producing structure but also as a key immunological interface, akin to a lymphatic organ within the brain [76,77]. It regulates immune surveillance by permitting selective trafficking of immune cells into the cerebrospinal fluid (CSF) [77] and expresses a wide range of cytokines, chemokines, and pattern-recognition receptors that can respond dynamically to systemic inflammation [77]. The CP of the fourth ventricle is the largest in the brain and lies in close anatomical proximity to the medulla and pons, regions central to autonomic and cardiorespiratory control [76]. In the context of Long COVID, inflammation of this structure could therefore exert disproportionate effects on brainstem function. Cytokines or immune cell activity within the fourth ventricle CP may diffuse into adjacent brainstem tissue or alter CSF composition, amplifying microglial activation and disrupting neural circuits critical for homeostasis [78]. Given its size, immune activity here may represent a major pathway linking persistent peripheral immune activation, such as macrophage-driven inflammation, leading to the neurological and autonomic symptoms characteristic of Long COVID [11,34].

A macrophage-centric model of Long COVID provides a coherent framework to understand the vast and varied symptomatology of the disease. Central to this hypothesis is the persistent presence of SARS-CoV-2 spike protein in immune cells such as CD16^+^ non-classical monocytes, even up to 15 months post-infection, suggesting ongoing antigenic stimulation and immune dysregulation [11]. This is further supported by imaging studies revealing hypermetabolic lymphadenopathy, particularly in axillary lymph nodes, persisting long after vaccination or infection, a sign of chronic macrophage and antigen-presenting cell activation [4,79]. These immune cells traffic spike antigen into diverse tissues including the gut, brainstem, and vascular endothelium. The resultant pathology depends not only on the tissue targeted but also on the epigenetic programming of monocytes and macrophages, which can lock them into a hyper-inflammatory phenotype, perpetuating symptoms even in the absence of replicating virus [12].

This model aligns with the observed mosaic pattern of disease presentation in Long COVID. For some individuals, persistent immune activation in the gut leads to symptoms resembling postural orthostatic tachycardia syndrome (POTS), fatigue, and gastrointestinal disturbances; for others, the primary dysfunction localizes to the brainstem and central nervous system, where microglial activation and immune signaling disrupt autonomic and cognitive function [74]. Neuroimaging has shown structural changes in the brainstem and hippocampus [43], and retinal microvascular abnormalities further confirm vascular involvement with CNS linkage [80]. The predominance of neurological symptoms such as brain fog, dizziness, and sleep disturbance likely reflects the vulnerability of the brainstem to circulating inflammatory mediators and spike-carrying immune cells [81]. The brainstem’s high density of serotoninergic and noradrenergic nuclei, along with its proximity to CSF drainage pathways, makes it particularly susceptible to persistent microglial activation—which may explain why neurological symptoms are among the most persistent and disabling in Long COVID [8]. In addition, a recent publication has highlighted that interferon gamma is suppressed which suggests that MAIT cells may be exhausted in some cases and lead to poor mucosal immunity [63].

This macrophage-centric model of brainstem dysfunction integrates structural, epigenetic, and functional data and offers a unifying framework for key Long COVID symptoms. It emphasizes that immune dysregulation, not classical autoimmunity, drives the chronic phase, with the brainstem macrophage axis acting as a central hub for symptom generation.

## 13. Conclusions

Figure 1 below summarizes the hypothesis described herein. Our hypothesis proposes that Long COVID is best understood as a disorder of persistent immune dysregulation, with chronic macrophage activation representing the fundamental underlying pathophysiology (Figure 1, center). Unlike transient post-viral syndromes, Long COVID involves a sustained innate immune response, particularly within monocyte-derived macrophages, driven by persistent spike protein (peripherally in MAIT cells and centrally in Microglial cells), epigenetic imprinting, and gut-related viral reservoirs. These macrophages are not merely activated temporarily but become epigenetically “trained” into a prolonged inflammatory state, as demonstrated by enduring histone acetylation markers such as H3K27acDNA Reprogram.

Table 1 below offers a summary of the proposed mechanisms, all of which were discussed and cited above, by which different macrophage phenotypes at their various sites of action might lead to the symptoms most commonly observed in Long COVID. The unresolving macrophage activation becomes the lens through which other immune disturbances—such as Treg dysfunction or MAIT cell hyper-reactivity—are not merely peripheral but actively fuel the fire. 

When regulatory controls fail, macrophages become unchecked producers of cytokines, tissue-damaging enzymes, and metabolic disruptors. Crucially, the symptoms of Long COVID are not random; they localize preferentially to tissues and organs with high resident or recruited macrophage activity. The brainstem, with its dense microglial (macrophage-like) population, explains the predominance of neurological symptoms including dysautonomia, fatigue, and breathlessness. Similarly, the gastrointestinal tract, rich in gut-associated macrophages—manifests symptoms linked to viral persistence, immune tolerance failure, and microbial imbalance.

Clinically, this model offers a compelling explanation for chest pain in Long COVID patients who present with normal coronary angiography or echocardiography. The pain may arise not from epicardial ischemia, but from aortic wall inflammation producing altered vascular compliance and heightened pulsatile stress. Recognition of this mechanism is critical, as it identifies a treatable target for vascular-directed therapies and underscores the need for advanced imaging modalities (PET/MRI) to detect active inflammation in symptomatic patients with possible aortitis-related symptoms.

This understanding leads naturally to a unifying mosaic theory of Long COVID. Individual symptom patterns arise from overlapping layers of immune dysfunction centered around macrophage activity, modified by pre-existing conditions. What emerges is a multi-dimensional disease model in which macrophage dysfunction not only explains the diversity of symptoms but also serves as a convergence point for viral, microbial, and autoimmune processes.

Recognizing macrophage activation as the axis of pathology provides clarity in the chaos. It offers a framework for personalized risk assessment, targeted intervention, and therapeutic recalibration—not just for long COVID, but potentially for a range of post-infectious and macrophage-centric diseases.

## Figures and Tables

**Figure 1 ijms-27-00313-f001:**
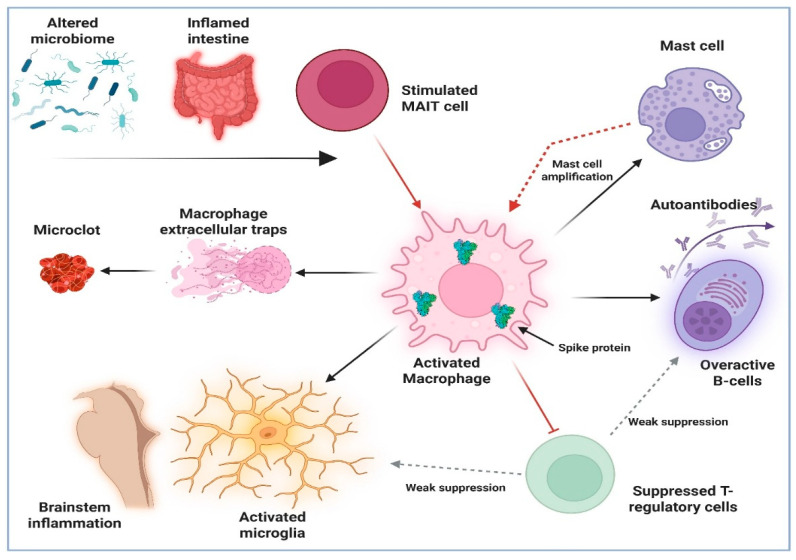
Summary of the Central Role of Macrophages in Long COVID hypothesized herein. SARS-CoV-2 infection is known to alter the gut microbiome [29], which together with increased intestinal permeability, can lead to activation of MAIT (Mucosal-associated invariant T) cells and promote chronic systemic inflammation. MAIT cells, by expressing high levels of IFN-g, TNF-a and IL-17, further stimulate monocyte-derived macrophages that were already activated by viral proteins, RNA or their remnants (center), into a hyper-activated state that is maintained in part by epigenetic mechanisms [6,12,37]. These hyperactivated macrophages can release macrophage extracellular traps (METs) [61] which, together with monocyte-derived tissue factor and cytokines, propagate a cycle of thromboinflammation [62]. They can also activate subsets of tissue-resident mast cells [65,66], promote B-cells to form autoantibodies found in Long COVID [34] and activate microglia in brainstem regions close to the choroid plexus, in areas where imaging studies have found evidence of neuroinflammation that correlates with the severity of symptoms [2,3]. It is proposed that both of these downstream effects may be amplified by impaired T-regulatory cell function, potentially weakened through exposure to viral spike protein [49,50]. The authors acknowledge that the links proposed above, while theorized on the basis of existing evidence from various sources cited, are still hypothetical and thus in need of confirmation through further research.

**Table 1 ijms-27-00313-t001:** Summary of Macrophage Phenotypes and Corresponding Symptoms in Long COVID.

Phenotype (Dominant Domain)	Primary Site of Macrophage/ Innate Immune Activation	Core Symptom Characteristics	Likely Abnormal Cytokines /Mediators	Pathophysiological Interpretation
Epipharyngeal–Vagal (ME-CFS–like)	Epipharyngeal mucosa, vagal afferents, nodose/jugular ganglia	Severe fatigue, post-exertional symptom exacerbation, POTS, palpitations, temperature dysregulation, unrefreshing sleep. Closest to MECFS	↑IL-6,↑TNF-α, ↑IL-1β,↑IFN-γ (low-grade), ↑CXCL8 (IL-8), ↓TGF-β (relative)	Chronic mucosal macrophage activation sustains vagal inflammation with afferent signaling and autonomic instability; cytokines act as neuromodulators rather than systemic inflammatory drivers
Gut–Immune	Intestinal lamina propria MAIT cells, GALT especially in ileum, mesenteric lymphatics	Abdominal pain, bloating, diarrhoea/constipation, food intolerance, post-prandial fatigue, brain fog	↑IL-6, ↑TNF-α, ↑IL-1β, ↑IL-18, ↑IL-23, ↑IL-17axis	Barrier dysfunction and microbial translocation maintain chronic innate MAIT immune priming, feeding systemic fatigue and cognitive symptoms through macrophage activation in brain and periphery
Choroid Plexus/Brainstem	Choroid plexus macrophages, CSF–brain interface, brainstem nuclei	Nausea, dizziness, vertigo, head pressure, headaches, sleep disturbance, central air hunger	↑IL-6, ↑IL-1β, ↑IFN-γ, ↑CCL2(MCP-1), ↑CXCL10 (IP-10)	Cytokine signaling at the CSF–brain interface primarily in fourth ventricle disrupts autonomic, vestibular, and respiratory patterning, despite normal structural imaging
Vascular–Endothelial	Endothelium, vasa vasorum, perivascular macrophages, circulating CD16^+^ monocytes	Exertional dyspnoea, chest tightness, exercise intolerance, palpitations	↑TNF-α, ↑IL-6, ↑IL-1β, ↑CCL2, ↑CX3CL1 (fractalkine), ↑VEGF, ↑platelet-activating mediators	Persistent monocyte–endothelial activation drives microvascular dysfunction and impaired oxygen delivery, distinct from parenchymal lung disease

## Data Availability

No new data were created or analyzed in this study. Data sharing is not applicable to this article.

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
