# Peer review of "The Central Role of Macrophages in Long COVID Pathophysiology"

_ijms, 2025, doi:10.3390/ijms27010313_

Round 1

Reviewer 1 Report

Comments and Suggestions for Authors

The work by McMillan et al., provides a thorough review and perspective of the role of immune cells and especially macrophages in explaining the different aspects of the pathophysiology of long COVID. While the role of macrophages in COVID is well established, a unifying model centered on macrophages that could explain the myriad of long COVID symptoms was largely missing and this expert opinion effectively addresses this gap. Overall, the manuscript is well-written and interesting to read. I have prepared a few suggestions that may help the authors in providing a revised version of their work for further consideration.

1.The authors could more thoroughly embrace in their unifying model results from recent omics methodologies such as single cell RNA-seq and epigenomics studies to further support the important roles of macrophages and other immune cells in long covid pathophysiology. Some examples include the following: (PMID: 37597510, PMID: 34108657, PMID: 39018367, PMID: 38744274, PMID: 38212464, PMID: 35385861, PMID: 38236961). This may allow readers to derive a more mechanistic understanding of their proposed model regarding the role of altered macrophage subset transcriptional programmes and DNA regulatory elements in different milieus, as well as their effect on the proteome and the metabolic landscape.

  1. Further references describing the role of macrophages as reservoirs for lasting infection could be mentioned (PMID: 35385861).
  2. The authors could expand more on the role of epigenetic memory in explaining long covid, especially the modulation of myeloid progenitors that could sustain a long-standing memory (PMID: 37597510). Evidence for such a progenitor memory also comes from other settings (PMID: 39153479) and may even be manipulated at will in the future (PMID: 41162785, PMID: 40930103).

4.References 7,10,11 are probably the same, please correct accordingly

5.Please report all references in the same style, including DOI numbers.

6.Please correct line 95

Author Response

Reviewer 1

We thank the reviewer for the laudatory comments and the thoughtful and constructive criticisms. The following is a point-by-point response and listing of specific revisions:

Critique: “The authors could more thoroughly embrace in their unifying model results from recent omics methodologies…”

Response: This has been added as suggested (new ref#9)

Critique: “Further references describing the role of macrophages as reservoirs for lasting infection could be mentioned”

Response: This was highlighted with the Patterson papers.

Critique: “The authors could expand more on the role of epigenetic memory in explaining long covid, especially the modulation of myeloid progenitors…”

Response: There has been an addition of involvement of myeloid suppressor progenitors as a driver for lymphopenia (refs ~18)

Critique: “References 7,10,11 are probably the same, please correct accordingly”

Response: All references were checked and duplicates were removed.

Critique: “Please report all references in the same style, including DOI numbers.”

Response: All references were checked and DOI numbers added.

Critique: “Please correct line 95”

Response: This line has been corrected

Reviewer 2 Report

Comments and Suggestions for Authors

Regarding the manuscript by McMillan et al., entitled "The Central Role of Macrophages in Long COVID Pathophysiology", I have the following concerns.

  1. The manuscript relies heavily on the concept of "trained immunity" through H3K27ac (histone 3 lysine 27 acetylation) to explain chronic inflammation. The authors need to clearly distinguish between the acute cytokine storm (similar to M1) and the chronic phenotype of "trained" macrophages (which is distinct and often characterized by a specific, lower-grade, and sustained inflammatory profile). Furthermore, the relationship between MAIT cells and macrophages requires greater mechanistic precision based on current literature.
  2. The authors need to clarify how the H3K27ac marks result in an "open chromatin" state at the promoters of inflammatory genes. It is possible that these cells do not produce massive amounts of cytokines at baseline (which would resemble acute sepsis), but rather react disproportionately to minor triggers (e.g., intestinal microbial translocation, which is indicated).
  3. Is the pathology due to excessive MAIT activity (cytotoxicity) or dysfunctional MAIT activity (failure in tissue repair)?
  4. The information regarding the "M1 vs. M2" dichotomy needs to be expanded, as it is too simplistic for PASC. "Trained" macrophages often exhibit a hybrid phenotype.
  5. The authors should expand on the information regarding enhancers; for example, H3K27ac is the hallmark of active enhancers in trained immunity. However, H3K4me3 (promoter preparation) is also crucial in SARS-CoV-2 spike-induced training, enabling faster transcription upon re-exposure.
  6. It is suggested that "long-lasting histone acetylation" causes a "sustained innate immune response," however. H3K27ac creates chromatin accessibility. It acts as a "memory" awaiting a trigger. The manuscript implies a constant "on" switch; but the literature suggests a "trigger-sensitive" switch. This would explain why patients with PASC experience a collapse after exercise/stress (the trigger) rather than having constant cytokine levels similar to those seen in sepsis.
  7. It is suggested to include a figure showing the interaction between the gut, MAIT cells, and systemic macrophage training.

Author Response

Reviewer 2:

We thank the reviewer for the thoughtful and constructive criticisms. The following is a point-by-point response and listing of specific revisions:

Critique: The manuscript relies heavily on the concept of "trained immunity" through H3K27ac (histone 3 lysine 27 acetylation) to explain chronic inflammation. The authors need to clearly distinguish between the acute cytokine storm (similar to M1) and the chronic phenotype of "trained" macrophages (which is distinct and often characterized by a specific, lower-grade, and sustained inflammatory profile).

Response: M1 vs “trained” macrophages are now discussed in the section “MAIT Cells, Gut Barrier…”

Critique: Is the pathology due to excessive MAIT activity (cytotoxicity) or dysfunctional MAIT activity (failure in tissue repair)?

Response: We agree this is a critical issue, but discuss that insufficient autopsy specimens have been examined to make such determinations.

Critique: The information regarding the "M1 vs. M2" dichotomy needs to be expanded, as it is too simplistic for PASC. "Trained" macrophages often exhibit a hybrid phenotype.

Response: We now discuss M1 vs trained macrophages in the section “MAIT Cells, Gut Barrier…”

Critique: It is suggested that "long-lasting histone acetylation" causes a "sustained innate immune response," however. H3K27ac creates chromatin accessibility. It acts as a "memory" awaiting a trigger. The manuscript implies a constant "on" switch; but the literature suggests a "trigger-sensitive" switch.

Response: We thank the reviewer for this interesting and relevant observation.

Critique: It is suggested to include a figure showing the interaction between the gut, MAIT cells, and systemic macrophage training.

Response: We now include the new Figure 1.

Reviewer 3 Report

Comments and Suggestions for Authors

In their manuscript, McMillan et al. raised a complex and pressing issue in modern medicine: determining the role of macrophages in the pathogenesis of long-term COVID-19. It's worth noting that the primary challenge in studying this disease is the need to conduct differential diagnosis with a wide range of other conditions. Furthermore, long-term COVID-19 includes a wide range of nonspecific syndromes, complicating the study of its pathogenesis. In their work, the authors proposed an immune-centric model for the pathogenesis of long-term COVID-19, focusing on the monocyte-macrophage component of the immune system. The idea itself appears quite interesting and relevant, but it lacks specific evidence regarding the relationship between macrophages and various disease manifestations. The text is characterized by numerous semantic repetitions and inconsistencies between the literature sources and the facts presented in the manuscript. The manuscript also requires supplementing with figures and tables to provide a clearer understanding of the role of macrophages in the pathogenesis of long-term COVID-19. The manuscript requires significant revision without losing its focus.

Suggestions:

  1. Line 25. Explain what the authors mean by the term "therapeutic recalibration."
  2. Lines 53–54. Are there differences in immune cell reprogramming after vaccination and after COVID-19? How often do vaccinated people experience the same symptoms as patients with non-organ damaged-related syndromes after infection? Do vaccinated subjects with and without symptoms have the same mechanisms of chronic macrophage activation? Explain in the text.
  3. Lines 88–90. How exactly do IL-1β and IL-18 enhance chronic activation of the immune system? Does this lead to immune dysregulation only or to immune dysregulation and target organ damage? Please explain the mechanisms in the text of the manuscript.
  4. Line 95. Please correct the mistake "persistent".
  5. Lines 101–102. How exactly do immune cells suppress adaptive T cell responses? Describe the mechanisms in the text of the manuscript.
  6. Section "Macrophage Activation: The Common Pathophysiological Axis in Severe and Long COVID." Perhaps adding a discussion of the role of macrophages in lymphopenia and germinal center depletion in severe COVID-19 would improve this manuscript.
  7. Section "Macrophage Activation: The Common Pathophysiological Axis in Severe and Long COVID." It is worth providing statistics on the development of long COVID-19 in patients after severe and mild infection.
  8. Lines 132–135. The reference does not correspond to the study described in the text. It is worthwhile to discuss this study in more detail and address the following questions in the text. Did the patients report any complaints? What was the basis for the PET scan? Was the differential diagnosis considered with Takayasu's vasculitis?
  9. Lines 158–160. Describe the mechanism of MAIT cell activation by coronavirus proteins.
  10. Lines 166–168. How exactly do MAIT cells leave the mucosa and mucosa-associated lymphoid tissue? What do they interact with? How far can they migrate? Do they continuously secrete cytokines? Explain in the text of the manuscript, citing specific research references.
  11. Lines 186–189. Can an already activated proinflammatory macrophage change its phenotype to a regulatory one in vivo?
  12. Lines 189–190. Please describe the mechanism of fibrosis induced by activated macrophages in more detail in the text of the manuscript.
  13. Lines 193–195. The reference is incorrect.
  14. Lines 195–198. The reference is incorrect. Please double-check all references in the manuscript.
  15. Lines 203–206. Perhaps a description of experimental drugs that block IFN-γ signaling and epigenetic reprogramming of macrophages, as well as potential targets for them, would improve this manuscript.
  16. Lines 216–218. Incorrect reference. It is also worth mentioning specific subpopulation phenotypes in the text of the manuscript due to the wide range of Tem and Trm gating strategies.
  17. Lines 221–223. The study reference is required.
  18. Lines 235–239. The references do not match the information provided in the manuscript text. This should be corrected.
  19. Lines 245–248. Do IL-3 or IL-3-producing CD8+ T cells penetrate the blood-brain barrier to activate microglia? Describe the mechanism in the manuscript text.
  20. Lines 270–273. Is there a link between prior IBD and spike protein uptake by intestinal mucosal CD8+ T cells? Explain in the text.
  21. The section "Regulatory T Cell Dysfunction as a Catalyst for Macrophage Overactivation in Long COVID" requires a detailed discussion of the mechanisms of interaction between regulatory T cells and macrophages.
  22. Line 339. How does PD-L1 and TIM-3 expression impair mucosal healing and promote systemic inflammation?
  23. Lines 376–378. The semantic context of this sentence is repeated numerous times in the manuscript. Please rephrase these sentences.
  24. Line 431–433. The relationship between macrophages and mast cells needs to be described in more detail.
  25. Line 433–436. Provide specific examples of the use of mast cell stabilizers, antihistamines, and leukotriene receptor antagonists in long-term COVID-19 in the manuscript.
  26. Line 468–471. This sentence is repeated numerous times in the manuscript. It needs to be rephrased.
  27. Line 482–485. It's unclear what exactly might diffuse into adjacent brainstem tissue and alter the composition of the CSF. The sentences need to be reformulated for reader-friendly purposes.
  28. Lines 517–520. What approaches to vascular-targeted therapy do the authors propose? It is necessary to clarify the purpose of using PET/MRI: for differential diagnosis or as part of assessing the prevalence of inflammatory changes in the vascular wall in already verified long-COVID-19?
  29. Line 530. Provide examples in the text of the manuscript of macrophage-centric diseases that developed after COVID-19.
  30. A graphical representation of the immune-centric model of the long-term course of COVID-19, the central element of which is macrophage activation, could improve this manuscript.
  31. A generated table of the phenotypes and symptoms of long-COVID-19, as well as the corresponding macrophage characteristics, could improve this manuscript.

Author Response

Reviewer 3:

We thank the reviewer for the thoughtful and constructive criticisms. We appreciate the many suggestions for revision, and in particular have attempted to revise the manuscript to achieve “significant revision without losing its focus”. To that end, a new Figure (Figure1) was assembled to provide a pictorial summary.

The following is a point-by-point response and listing of specific revisions:

Critique: “Line 25. Explain what the authors mean by the term "therapeutic recalibration."

Response: This is referring to bringing a previous inflammatory process back into balance.

Critique: “Lines 53–54. Are there differences in immune cell reprogramming after vaccination and after COVID-19?”

Response: This is unknown, but our premise is that the epigenetic reprogramming can occur from any source of spike protein.

Critique: “Lines 88–90. How exactly do IL-1β and IL-18 enhance chronic activation of the immune system? Does this lead to immune dysregulation only or to immune dysregulation and target organ damage? Please explain the mechanisms”

Response: This is demonstrated in the Patterson paper with elevated interferon in long Covid linked to non-classical macrophages (Section – Macrophage subsets) and discussed the addition of multi-omic studies in the Introduction.

Critique: “Line 95. Please correct the mistake "persistent".

Response: Corrected

Critique: Lines 101–102. How exactly do immune cells suppress adaptive T cell responses? Describe the mechanisms in the text

Response: This is now discussed with the myeloid suppressor cell line (Section – Macrophage Activation).

Critique: “Section "Macrophage Activation: The Common Pathophysiological Axis in Severe and Long COVID." Perhaps adding a discussion of the role of macrophage”

Response: The paper is aiming to be a broad overview of long covid and as such, balances going into too much detail on any aspect.

Critique: “It is worth providing statistics on the development of long COVID…”

Response: Provided in the introduction

Critique: “Lines 132–135. The reference does not correspond to the study described in the text. It is worthwhile to discuss this study in more detail”

Response: Reference corrected. Our aim is to show how multiple patterns of symptoms are connected by inflammation from one main cell type. Each one of these areas might easily become a paper in the future.

Critique: “Lines 158–160. Describe the mechanism of MAIT cell activation by coronavirus proteins”

Response: Discussion was added in section of MAIT cells and Gut Barrier.

Critique: “Lines 166–168. How exactly do MAIT cells leave the mucosa and mucosa-associated lymphoid tissue?”

Response: MAIT cells will normally circulate through the body; exact mechanisms are still being defined.

Critique: “Lines 186–189. Can an already activated proinflammatory macrophage change its phenotype to a regulatory one in vivo?”

Response: This is the premise on how long covid can eventually be resolved - by removing the immune drivers.

Critique: “Lines 189–190. Please describe the mechanism of fibrosis induced by activated macrophages in more detail in the text of the manuscript.”

Response: Paper 5 was added to support this point

Critique: “Lines 193–195. The reference is incorrect”

Response: Now corrected

Critique: “Lines 195–198. The reference is incorrect.”

Response: Now corrected

Critique: “Lines 203–206. Perhaps a description of experimental drugs that block IFN-γ signaling and epigenetic reprogramming of macrophages, as well as potential targets for them, would improve this manuscript.”

Response: This paper represents a hypothesis and so specific mention of any therapies was avoided.

Critique: “Lines 216–218. Incorrect reference. It is also worth mentioning specific subpopulation phenotypes in the text of the manuscript due to the wide range of Tem and Trm gating strategies”

Response: Thank you for the input.

Critique: “Lines 221–223. The study reference is required.”

Response: Paper 30 added as requested

Critique: “Lines 235–239. The references do not match the information provided in the manuscript text. This should be corrected.”

Response: Now corrected

Critique: “Lines 245–248. Do IL-3 or IL-3-producing CD8+ T cells penetrate the blood-brain barrier to activate microglia? Describe the mechanism in the manuscript”

Response: The point about involvement of the 4th ventricle and the proximity of immune cells to the circumventricular organs allows cytokines to more easily cross the blood brain barrier.

Critique: “Lines 270–273. Is there a link between prior IBD and spike protein uptake by intestinal mucosal CD8+ T cells? Explain in the text.”

Response: This was covered in more detail in our previous paper (Reference 26)

Critique: “The section "Regulatory T Cell Dysfunction as a Catalyst for Macrophage Overactivation in Long COVID" requires a detailed discussion of the mechanisms of interaction between regulatory T cells and macrophages”

Response: This section can also be a whole paper in itself and could not be adequately covered. It helps the reader to understand the overview of the condition and overlapping links.

Critique: “Line 339. How does PD-L1 and TIM-3 expression impair mucosal healing and promote systemic inflammation?”

Response: PD-L1 while protecting a specific cell with spike protein could allow surrounding inflammatory processes potentially leading to fibrosis.

Critique: “Lines 376–378. The semantic context of this sentence is repeated numerous times in the manuscript. Please rephrase these sentences”

Response: Rephrasing completed

Critique: “Line 431–433. The relationship between macrophages and mast cells needs to be described in more detail.”

Response: This point can become too large for an overview paper as mast cells represent only a part of the picture.

Critique: “Line 433–436. Provide specific examples of the use of mast cell stabilizers, antihistamines, and leukotriene receptor antagonists in long-term COVID-19 in the manuscript.”

Response: Paper 55 in references touched on the point.

Critique: “Line 468–471. This sentence is repeated numerous times in the manuscript. It needs to be rephrased.”

Response: Now reformmated

Critique: ” Line 482–485. It's unclear what exactly might diffuse into adjacent brainstem tissue and alter the composition of the CSF. The sentences need to be reformulated for reader-friendly purposes.

Response: Now reformmated

Critique: “Lines 517–520. What approaches to vascular-targeted therapy do the authors propose? It is necessary to clarify the purpose of using PET/MRI: for differential diagnosis or as part of assessing the prevalence of inflammatory changes in the vascular wall in already verified long-COVID-19?”

Response: The connection of aortic inflammation and long Covid symptoms represents a unique concept and as such research is limited.

Critique: “Line 530. Provide examples in the text of the manuscript of macrophage-centric diseases that developed after COVID-19.”

Response: This is referring to Patterson and the specific elevation of cytokines that support the premise of macrophage overactivity.

Critique: “A graphical representation of the immune-centric model of the long-term course of COVID-19, the central element of which is macrophage activation, could improve this manuscript.”

Response: This was added as Figure 1

Critique: “A generated table of the phenotypes and symptoms of long-COVID-19, as well as the corresponding macrophage characteristics, could improve this manuscript.”

Response: Thank you for the suggestion but at present it would be beyond the scope of this paper.

Round 2

Reviewer 1 Report

Comments and Suggestions for Authors

The authors have responded adequately to most of comments and the quality of the manuscript has been improved.

Author Response

We thank the reviewer for the positive comments

Reviewer 2 Report

Comments and Suggestions for Authors

I think the authors have answered all the observations, however Turnitin reveals that 44% was generated by AI, so the authors should correct this.

Author Response

We thank the reviewer for the constructive comments on the use of AI tools. With all due respect, we counterargue that all the authors take issues of originality and academic integrity very seriously. This paper was conceived, structured, and written by the authors based on their own scientific expertise, interpretation of the literature, and previously published work. No generative AI tool was used to create original scientific ideas, fabricate data, or substitute for scholarly judgment. The depth and complexity of this research paper could not easily be achieved without using all tools available.

Automated similarity or AI-detection tools are known to have limitations, particularly for review articles that employ standardized scientific language, mechanistic descriptions, and commonly used immunological terminology. 

Nevertheless, in response to the reviewer’s comment, the authors have carefully revised the manuscript to improve stylistic variability, clarity, and authorial voice, while preserving scientific accuracy. Any sections with repetitive or formulaic phrasing have been rewritten. The revised manuscript reflects the authors’ original scholarship and interpretation of the literature.

Reviewer 3 Report

Comments and Suggestions for Authors

McMillan et al. did a great job editing their manuscript. After making these edits, the manuscript has significantly improved, but important corrections are still needed. Despite the clearly stated purpose of the literature review, there are deviations from it in some sections of the manuscript. Furthermore, many immune mechanisms are not described in detail, although they should be an important part of the literature review. The relationship between monocyte/macrophage activation and symptoms in long-term COVID-19 is unclear. One image was added, which significantly improved the manuscript. However, a tabular presentation of the review's summary data is missing. This would make the review more reader-friendly. Currently, the manuscript requires significant revision.

Suggestions:

  1. "Response: The paper aims to be a broad overview of long COVID, and as such, balances going into too much detail on any aspect." - This response differs from the objective stated in the introduction to this review. Please correct the objective of the review or revise the suggestion "Section "Macrophage Activation: The Common Pathophysiological Axis in Severe and Long COVID." Perhaps adding a discussion of the role of macrophage."
  2. The Introduction section does not provide statistics on the development of Long-COVID-19 in patients with mild and severe COVID-19.
  3. The precise mechanisms of MAIT cell circulation are not fully understood, but some have been well described in the literature. Please discuss these in the text of the manuscript.
  4. If an already activated proinflammatory macrophage can change its phenotype to a regulatory one and this is a prerequisite for the ultimate cure of COVID-19, then this deserves a detailed discussion in the manuscript text, with references provided.
  5. The mechanism of fibrosis induced by activated macrophages is still missing from the manuscript text.
  6. Lines 232–233. The Tem and Trm phenotypes should be mentioned in the manuscript text due to the wide range of Tem and Trm gating strategies.
  7. Line 359. A brief explanation in the manuscript text is needed of how PD-L1 and TIM-3 expression impair mucosal healing and promote systemic inflammation.
  8. Lines 569–571. This proposal remains unclear. The authors mention the need to use PET/MRI, but do not provide specific indications for this. How can subclinical inflammation cause symptoms in patients, given that it is subclinical? What specific symptoms are we talking about? If research in this area is limited, why do the authors mention the need for PET/MRI? Please reconsider the wording of this sentence. Could the sentence be more "lenient" or could the purpose of using PET/MRI be clarified: for differential diagnosis or as part of assessing the prevalence of inflammatory changes in the vascular wall in patients with confirmed long-term COVID-19?
  9. If I understand correctly, the authors' goal is to demonstrate the relationship between monocytes/macrophages and symptoms of long COVID-19, with an emphasis on cytokine profiling and epigenetic reprogramming (Lines 69–72). Why, then, is a summary table describing macrophage phenotypes and corresponding symptoms beyond the scope of this manuscript?

Author Response

Reviewer 3 Responses:

We thank the reviewer for the comments and suggestions. The following are point-by-point responses:

  1. Critique: The paper aims to be a broad overview of long COVID, and as such, balances going into too much detail on any aspect." - This response differs from the objective stated in the introduction to this review. Please correct the objective of the review or revise the suggestion "Section "Macrophage Activation: The Common Pathophysiological Axis in Severe and Long COVID." Perhaps adding a discussion of the role of macrophage."

Response: Thank you for this recommendation. We have retained the original wording in the Introduction “This paper proposes an immune-centric framework for Long COVID…” but chose instead to revise the suggestion to read “Section “Macrophage Activation: A Common Pathophysiological Axis in Severe and Long COVID?”, which, posed now as a question, invites the reader to consider our hypotheses for themselves. With due respect to the reviewer, we counter that the hypothesis described in the new Figure1 and Legend already discuss our proposed roles of the macrophage…which we clearly acknowledge are still hypothetical. A restatement of this was added to the legend.

  1. Critique: The Introduction section does not provide statistics on the development of Long-COVID-19 in patients with mild and severe COVID-19.

Response: Another citation (9) with comments has been added in the introduction.

  1. Critique: The precise mechanisms of MAIT cell circulation are not fully understood, but some have been well described in the literature. Please discuss these in the text of the manuscript.

Response: A paragraph with citations has been added to the MAIT Cells, Gut Barrier Dysfunction section.

  1. Critique: If an already activated proinflammatory macrophage can change its phenotype to a regulatory one and this is a prerequisite for the ultimate cure of COVID-19, then this deserves a detailed discussion in the manuscript text, with references provided.

Response :A paragraph and citations on the link to fibrosis was added in Aortic inflammation and Vasa Vasorum Pathology

  1. Critique: The mechanism of fibrosis induced by activated macrophages is still missing from the manuscript text.

Response: A paragraph and citations on the link to fibrosis was added in Aortic inflammation and Vasa Vasorum Pathology

  1. Critique: Lines 232–233. The Tem and Trm phenotypes should be mentioned in the manuscript text due to the wide range of Tem and Trm gating strategies.

Response: A short sentence was added to the CD8 T-Cell Hyperactivation section

  1. Critique: Line 359. A brief explanation in the manuscript text is needed of how PD-L1 and TIM-3 expression impair mucosal healing and promote systemic inflammation.

Response: Adjustments were made to section Interferon-Driven MAIT Cell Hyperactivation

  1. Critique: Lines 569–571. This proposal remains unclear. The authors mention the need to use PET/MRI, but do not provide specific indications for this. How can subclinical inflammation cause symptoms in patients, given that it is subclinical? What specific symptoms are we talking about? If research in this area is limited, why do the authors mention the need for PET/MRI? Please reconsider the wording of this sentence. Could the sentence be more "lenient" or could the purpose of using PET/MRI be clarified: for differential diagnosis or as part of assessing the prevalence of inflammatory changes in the vascular wall in patients with confirmed long-term COVID-19?

Response: We appreciate the point and the sentence has been adjusted.

  1. Critique: If I understand correctly, the authors' goal is to demonstrate the relationship between monocytes/macrophages and symptoms of long COVID-19, with an emphasis on cytokine profiling and epigenetic reprogramming (Lines 69–72). Why, then, is a summary table describing macrophage phenotypes and corresponding symptoms beyond the scope of this manuscript?

Response: A new Table 1 was added with a sentence describing the intent of the table now in the Conclusion.

Round 3

Reviewer 3 Report

Comments and Suggestions for Authors

The authors have done a tremendous work of editing and correcting inaccuracies in the manuscript. Now the information in the manuscript is presented comprehensively, and I can recommend it for publication.

Author Response

Thank you, the revisions have now been made.